# *Arabidopsis* G-Protein β Subunit AGB1 Negatively Regulates DNA Binding of MYB62, a Suppressor in the Gibberellin Pathway

**DOI:** 10.3390/ijms22158270

**Published:** 2021-07-31

**Authors:** Xin Qi, Wensi Tang, Weiwei Li, Zhang He, Weiya Xu, Zhijin Fan, Yongbin Zhou, Chunxiao Wang, Zhaoshi Xu, Jun Chen, Shiqin Gao, Youzhi Ma, Ming Chen

**Affiliations:** 1Institute of Crop Sciences, Chinese Academy of Agricultural Sciences (CAAS)/National Key Facility for Crop Gene Resources and Genetic Improvement, Key Laboratory of Biology and Genetic Improvement of Triticeae Crops, Ministry of Agriculture, Beijing 100081, China; 13121253699@163.com (X.Q.); tang_wensi@yeah.net (W.T.); hz384298890@gmail.com (Z.H.); 15651912225@163.com (W.X.); zhouyongbin@caas.cn (Y.Z.); wcx51619@163.com (C.W.); xuzhaoshi@caas.cn (Z.X.); chenjun01@caas.cn (J.C.); 2State Key Laboratory of Elemento-Organic Chemistry, College of Chemistry, Nankai University, Tianjin 300071, China; fanzj@nankai.edu.cn; 3Beijing Advanced Innovation Center for Food Nutrition and Human Health, Beijing Technology & Business University (BTBU), Beijing 100048, China; liweiwei.0304@163.com; 4Beijing Engineering Research Center for Hybrid Wheat, Beijing Academy of Agriculture and Forestry Sciences, Beijing 100097, China; gshiq@126.com

**Keywords:** *Arabidopsis*, GA signaling, *AGB1*, *MYB62*, protein interaction

## Abstract

Plant G proteins are versatile components of transmembrane signaling transduction pathways. The deficient mutant of heterotrimeric G protein leads to defects in plant growth and development, suggesting that it regulates the GA pathway in *Arabidopsis*. However, the molecular mechanism of G protein regulation of the GA pathway is not understood in plants. In this study, two G protein β subunit (*AGB1*) mutants, *agb1-2* and *N692967*, were dwarfed after exogenous application of GA_3_. AGB1 interacts with the DNA-binding domain MYB62, a GA pathway suppressor. Transgenic plants were obtained through overexpression of *MYB62* in two backgrounds including the wild-type (*MYB62/WT* *Col-0*) and *agb1* mutants (*MYB62/agb1*) in *Arabidopsis*. Genetic analysis showed that under GA_3_ treatment, the height of the transgenic plants *MYB62/WT* and *MYB62/agb1* was lower than that of WT. The height of *MYB62/agb1* plants was closer to *MYB62/WT* plants and higher than that of mutants *agb1-2* and *N692967*, suggesting that *MYB62* is downstream of *AGB1* in the GA pathway. qRT-PCR and competitive DNA binding assays indicated that MYB62 can bind MYB elements in the promoter of *GA2ox7*, a GA degradation gene, to activate *GA2ox7* transcription. AGB1 affected binding of MYB62 on the promoter of *GA2ox7*, thereby negatively regulating th eactivity of MYB62.

## 1. Introduction

The heterotrimeric G protein pathway is a conservative transmembrane signal transduction pathway found in both animals and plants [1,2]. In *Arabidopsis*, the heterotrimeric G protein is composed of an α subunit (*GPA1*), a β subunit (*AGB1*), and three γ subunits (*AGG1*, *AGG2*, and *AGG3*) [3]. During G protein signaling transduction, G-α binds to GDP and forms a coupling polymer with the G-β-γ dimer, leaving the G protein pathway in a resting state [4] (Anantharaman et al., 2011). When the G protein pathway is activated, the GTP-G-α monomer is dissociated from the G-β-γ dimer, while the GTP-G-α monomer and the G-β-γ dimer interact with a variety of downstream effectors to transmit signals for different cell and physiological functions [4] (Anantharaman et al., 2011). The heterotrimeric G proteins are involved in growth and developmental processes such as seed germination and seedling development [5] (Ullah et al., 2003), cell division and morphology [6] (Ullah et al., 2002), ion channel regulation [7] (Assmann and Yu, 2015), stomatal development [8] (Wang et al., 2011), and the response to environmental conditions such as phytohormones, sugar, ROS (reactive oxygen species), and light [9] (Li et al., 2012). Plants with mutated G protein complex components have altered morphology in their fruits, grain weight, roots, and leaves, and these mutants are sensitive to a variety of hormones, including IAA (auxin), GA (gibberellins), and BR (brassinosteroids) [2,5,10,11,12] (Ullah et al., 2003; Chen et al., 2004; Pandey et al., 2006; Chen, 2007; Urano et al., 2014).

The G-α subunit positively regulates the GA pathway by inhibiting the activity of SLR (slender rice), a negative regulator of GA signaling in rice [13] (Ueguchi-Tanka et al., 2000), and the *Arabidopsis* G-α mutant *gpa1* is less sensitive to GA [14] (Trusov et al., 2007). The G-β subunit is involved in a variety of signaling pathways in plants. In *Arabidopsis*, the interactions between AGB1 and ERECTA during silique development were the first evidence of the interaction between G proteins and receptor kinases [15] (Lease et al., 2001). We previously found that AGB1 is involved in regulating ABA and drought response by interacting with the protein kinase MPK6 and the transcription factor VIP1 [16] (Xu et al., 2015). We also found that AGB1 interacts with the transcription factor BBX21 to regulate photomorphogenesis in *Arabidopsis* [17] (Xu et al., 2017). In rice, suppression of the G-β subunit gene, *RGB1*, causes dwarfism and browning of the internodes and lamina joint regions [18] (Utsunomiya et al., 2011). In *Arabidopsis*, the G-β subunit gene mutant *agb1* has shorter mature plants than the wild-type (WT) [19] (Urano et al., 2016). These results suggest that the G-β subunit also regulates plant development through the GA pathway, but the specific mechanism is not clear.

GA functions directly in regulating plant growth and development as well as crop yield [20] (Singh et al., 2002). GA-related genes can be divided into two categories: either the pathways involved in GA synthesis or degradation, or the GA signaling transduction pathway. In plants, a variety of enzymes are involved in GA synthesis, while GGPP (geranylgeranyl pyrophosphate), a precursor of GA synthesis, is catalyzed by GGPS (geranylgeranyl pyrophosphate synthase) [21] (Lange et al., 2003). GGPP is further catalyzed by CPS (copalyl pyrophosphate synthase) and KS (ent-kaurene synthase), which forms kaurene [22] (Morrone et al., 2009). Additionally, KO (ent-kaurene oxidase) and KAO (ent-kaurenoic acid oxidase) have important roles in the GA synthesis intermediate GA_12_-aldehyde [23,24] (Davidson et al., 2003; Sakamoto et al., 2004). GA_12_-aldehyde is a branch point of GA, which hydroxylates to form GA_12_ and GA_53_ under the oxidation of *GA20ox1* (gibberellin 20-oxidase gene). GA_12_ and GA_53_ produce GA_9_ and GA_20_, while the final form of bioactive GA is catalyzed by *GA30ox1* [25] (Hedden et al., 2012).

In rice, the plant is dwarfed due to the mutation of *GA20ox*, a key gene for GA synthesis [26] (Ashikari et al., 2002). The deletion of *KAO* can lead to severe dwarfing in many plants, such as rice (*d35* mutant) [24] (Sakamoto et al., 2004), maize (*dwarf3*) [27] (Helliwell et al., 2001), pea (*na*) [23] (Davidson et al., 2003), and sunflower (*dwarf2*) [28] (Fambrini et al., 2011). The signal transduction of GA in plants is mediated by the receptor GID1, while DELLA is an inhibitor of the GA pathway via transcriptional regulation [29] (Ueguchi-Tanaka et al., 2005). GA promotes the formation of the GA–GID1–DELLA complex via conformational changes caused by GID1 binding, where the DELLA protein is then ubiquitinated and degraded by the 26S proteasome to open the GA pathway [30,31] (Murase et al., 2008; Shimada et al., 2008). In *Arabidopsis*, the transcription factor *MYB62* is involved in the GA pathway, and overexpression of *MYB62* results in a GA-deficient phenotype, which suggests that *MYB62* is a suppressor in the GA pathway [32] (Devaiah et al., 2009).

In this study, we found that plant heights in two G-β subunit mutants, *agb1-2* and *N692967*, were significantly lower than in the WT following GA_3_ treatment, suggesting that the function of *AGB1* in the GA pathway is similar to that of *GPA1* in *Arabidopsis*. We found that AGB1 regulates the GA pathway by negatively regulating the DNA binding of MYB62, a GA pathway suppressor on the promoter of the GA degradation gene *GA2ox7*. The G protein complex regulates the GA pathway through the *AGB1-MYB62-GA2ox7* pair in *Arabidopsis*.

## 2. Materials and Methods

### 2.1. Plant Materials and Growth Conditions

The *agb1* mutant *agb1-2* (CS6536) has been described by Ullah et al. (2003). In the other *agb1* mutant *N692967* (SALK_204268C), the T-DNA insert occurs in chr4 16,477,780 of the *Col-0* genome (Appendix A). The expression of *AGB1* in mutants and *MYB62* in transgenic plants was identified by qRT-PCR using gene-specific primers (Appendix A). Because we could not obtain *MYB62* mutants, we overexpressed the *MYB62* gene in different genetic backgrounds to obtain transgenic plants, including *MYB62* transgenic plants in a WT background—*MYB62:GFP/WT-8* and *MYB62:GFP/WT-10*—and *MYB62* transgenic plants in an *agb1* background—*MYB62:GFP/agb1-2-1*, *MYB62:GFP/agb1-2-4*. The response of *Arabidopsis* plants to GA_3_ treatment was observed at the seedling stage. Before planting *Arabidopsis* (*Col-0*) seeds on plates, the seeds were soaked with 10% sodium hypochlorite for 10 min and then washed 3 times with sterile distilled water. The sterilized *Arabidopsis* seeds were treated at 4 °C in the dark for 3 days and then germinated on 1/2 MS medium (0.8% agar) for 7 days. When *Arabidopsis* grew to the 4-leaf stage, seedlings were transplanted into 1/2 MS medium containing 1 μΜ GA_3_, 10 μΜ GA_3_, or 100 μΜ GA_3_ and grown at 24 °C in the light for 16 h then at 20 °C in darkness for 8 h (Appendix A). The growth state was observed and recorded every day. Differences were observed when the seedings were grown under GA_3_ treatment for 10 days.

### 2.2. Measurement of Gibberellin Content

Total GA content, including GA_1_, GA_2_, GA_3_, GA_4_, GA_7_, and GA_20_, was measured using an ELISA kit (Plant GA ELISA Kit, X-Y Biotechnology company) according to the manufacturer’s instructions. Plant samples (0.1 g) were ground, then dissolved in 900 μL of PBS buffer, fully mixed and centrifuged at 12,000 rpm for 10 min at 4 °C. All standards and samples were added in duplicate to Micro-ELISA strip plate wells (Eppendorf, Germany). The volume for a standard curve sample and a measured sample was 50 μL, and nothing was added to the blank well. A total of 100 μL of HRP-conjugate reagent was added to each well, then the wells were covered with tinfoil and incubated for 60 min at 37 °C. After washing 5 times, 50 μL of chromogen Solution A and 50 μL of chromogen Solution B were added to each well. The plate was incubated for 15 min at 37 °C and 50 μL of a stop solution was added to each well. The optical density at 450 nm was read using a microtiter plate within 15 min. The standard curve was generated by plotting the average OD_450_ obtained for each of the 6 standard concentrations on the vertical (*y*) axis versus the corresponding concentration on the horizontal (*x*) axis. The concentration of the sample was calculated according to the equation of the standard curve.

### 2.3. Construction of Plasmids and Transgenic Lines

The full length of *MYB62* (AT1G68320) was amplified using the primers 1302-MYB62-F and 1302-MYB62-R and inserted into the pCambia-1302 vector (Clontech, San Francisco, CA, USA) with GFP at the *C*-terminal end through the *BamHI* restriction enzyme cutting site. The constructs were transformed into wild-type (*Col-0*) and mutant *Arabidopsis agb1-2* at the flowering stage by *Agrobacterium tumefaciens* (GV3101)-mediated transformation [33] (Clough et al., 1998). The seeds of the T0 generation of transgenic *Arabidopsis* were sown on selective medium (MS medium with 40 mg/L hygromycin), and the seedings were transplanted into the soil in pots. The seeds of theT1 and T2 transgenic lines were further screened by hygromycin. More than 95% of the seeds of the T2 generation with hygromycin resistance were homozygous lines. The *MYB62:GFP/agb1-2* transgenic plants were identified using kanamycin and hygromycin for screening and identification. The phenotypes of homozygous lines were analyzed under GA_3_ treatment in the T3 generation.

### 2.4. Extraction of RNA and Analysis of Gene Expression

Seedlings of *MYB62* transgenic lines and the mutants *agb1-2* and *N692967*, and the roots and leaves of WT *Arabidopsis* were used for gene expression analysis. The total plant RNA was extracted using the Trizol method (Zhuangmeng Total RNA Extraction Kit), and the RNA was reverse-transcribed into cDNA using the TransScript One-Step gDNA removal and cDNA Synthesis SuperMix kit (TransGen Biotech, Beijing, China). We performed qRT-PCR using cDNA as the template and the primers in Appendix A according to the Real Master Mix (SYBR Green) kit (TransGen Biotech, Beijing, China). Gene expression was calculated using the 2^−ΔΔ*CT*^ method [34] (Livak & Schmittgen, 2001). Relative quantitative results were calculated through normalization based on the control gene *ACT2* (AT3G18780).

### 2.5. Subcellular Localization Analysis

The subcellular localization of MYB62 protein was completed in WT *Col-0 Arabidopsis* protoplasts. Protoplasts were prepared from the wild-type seedling leaves of *Arabidopsis* before bolting, according to the methods used in a previous study [35] (Yoo, Cho, & Sheen, 2007). The full-length coding sequences of *AGB1* and *MYB62* were amplified using gene-specific primers (Appendix A), and *AGB1* and *MYB62* were inserted into the vector 16318h-GFP to express the fused proteins GFP-AGB1 and GFP-MYB62, respectively. The vectors 16318-AGB1-GFP and 16318-MYB62-GFP were then separately transformed into protoplasts. A confocal laser scanning microscope (LSM700, Zeiss, Yena, Germany) was used to observe the experimental results.

### 2.6. Yeast Two-Hybrid Assay

To analyze the interaction between AGB1 and MYB62, *AGB1* and *MYB62* were, respectively, inserted into the pGADT7 vector and the pGBKT7 vector in the *EcoR*1 and *BamH*1 restriction enzyme cutting sites. We prepared the yeast cells and completed vector transformation according to the manufacturer’s instructions (TaKaRa, Tokyo, Japan). The transformants were selected on a synthetic dextrose (SD) medium lacking leucine and tryptophan (SD/-Leu/-Trp). The yeast transformants from the SD (-Leu/-Trp) were then streaked onto a solid SD (-Leu/-Trp/-His/-Ade) medium, with or without 40 μg/mL X-α-gal, to observe and photograph their growth.

### 2.7. Bimolecular Fluorescence Complementation (BiFC) Assay

AGB1 and MYB62 were, respectively, fused to the N- and C-termini of the luciferase reporter gene LUC, while the constructed vector was transformed into a strain of *A. tumefaciens*, GV3101. The *A. tumefaciens* samples transformed with nLUC-AGB1 and MYB62-cLUC were selected and the OD value of *A. tumefaciens* was adjusted to 0.8 with the infection solution (10 mM MES, 150 μM AS, 10 mM MgCl_2_ 6H_2_O). The *A. tumefaciens* samples transformed using the control, *nLUC-AGB1*, and *MYB62-cLUC* were mixed as pairs of *nLUC* and *cLUC*, *nLUC-AGB1* and *cLUC*, *nLUC* and *MYB62-cLUC*, and *nLUC-AGB1* and *MYB62-cLUC*, then injected into *Nicotiana benthamiana* leaves. Before analyzing LUC activity, *N. benthamiana* was cultured in darkness for 72 h. We performed 3 biological replicates for each combination of nLUC- and cLUC-infected tobacco leaves as a control.

### 2.8. Pull-Down Assays

*MYB62* was inserted into the pMAL-c2x (MBP-Tag) vector to express MBP-labeled fusion proteins (Takara, Japan), and *AGB1* was inserted into the pGEX4T-1 (GST-Tag) vector to express GST-labeled fusion proteins (Takara, Tokyo, Japan). The vectors pMAL-c2x-MYB62 and pGEX4T-1-AGB1 were then introduced into *Escherichia coli BL21* (DE3). The expression of GST and MBP fusion proteins was induced by isopropylthio-β-galactoside (IPTG) and expressed at 16 °C for a minimum of 16 h. The fusion proteins of GST-AGB1 and MBP-MYB62 were purified by glutathione-agarose 4B (GE Healthcare, Stockholm, Sweden) beads and MBP-agarose gel, according to the instructions of the manufacturer. In total, 50 µL of each recombinant fusion protein was mixed with 1 mL of the binding buffer (40 mM HEPES, 10 mM KCl, 0.4 M sucrose, 3 mM MgCl_2_ 6H_2_O, 1 mM EDTA, 1 mM DTT) in the pull-down assay. After eluting the MYB62 protein from the MBP-agarose gel with a 10 μΜ maltose solution, the MYB62-MBP protein and the GST-AGB1 protein were incubated in a pull-down buffer for approximately 8 h at 4 °C. They were then centrifuged at 2000× *g* for 1 min and washed 5 times at 4 °C with a 1× PBS buffer (pH 7.4). The particles containing binding proteins were then boiled in a 1× PBS buffer, and the released proteins were separated with 10% SDS-PAGE. The antibodies MBP-Tag and GST Tag were analyzed using a Western blot analysis (Abcam, Cambridge, UK).

### 2.9. Transcriptional Activation Experiment in Yeast

The transcription activation experiment was performed according to the methods used in previous research [36] (Yamaji et al., 2009). In order to further explore whether *AGB1* affects the transcriptional activation of *MYB62*, we carried out transcriptional activation experiments in yeast cells (Figure 5B). *MYB62* was inserted into pBridge vector to construct pBridge-MYB62, and *MYB62* was fused with the binding domain of the GAL4 transcription factor (GAL4-BD), which can bind target sequences upstream of the reporter genes (histidine-deficient reporter gene) in yeast chromosomes. When *MYB62* was inserted into pBridge, the reporter gene could be activated depending on the transcriptional activation activity of *MYB62* (Figure 5B). The yeast transformed with the pBridge-*MYB62* vector could grow on the screening medium. When *AGB1* was inserted into another expression cassette of the pBridge-MYB62 vector to make pBridge-MYB62-AGB1, the effect of AGB1 on the transcriptional activation of *MYB62* could be detected (Figure 5B). These constructed bodies were then introduced into the yeast reporter strain AH109 (Yeast Protocols Handbook; TaKaRa, Japan). The transformed yeast cells were selected on the selective medium (SD/-Trp, SD/-Trp-Ade, SD/-Trp-His-Ade) and the growth status was recorded.

### 2.10. LUC Assay of MYB62 in Tobacco (N. benthamiana)

*MYB62* was inserted into the pCambia-1302 vector as an effector vector, and the 2000 bp promoter sequences of the downstream genes, including *GA2ox7*, were fused into the pGreenII 0800-LUC vector as a reporter vector. The reporter gene and the effector gene were introduced into GV3101, a strain of *A. tumefaciens*, and then injected into tobacco leaves. The activity of *LUC* was observed after 72 h of growth. Each sample was injected into 8 tobacco leaves, with 3 biological replicates performed for each.

### 2.11. Electrophoretic Mobility Shift Assay (EMSA)

MYB62-MBP and AGB1-GST proteins were induced by IPTG in *E. coli BL21* (DE3). The fusion proteins of GST-AGB1 and MBP-MYB62 were purified by glutathione-agarose 4B (GE Healthcare, North Richland Hills, TX, USA) beads and MBP-agarose gel, according to the instructions of the manufacturer. The synthetic oligonucleotide probe was synthesized by the ShengGong Biotech Company. The LightShift chemiluminescence EMSA kit (Thermo Science, Waltham, MA, USA) was used for EMSA. The biotin-labeled probe was incubated for 30 min at room temperature in a binding buffer (2.5% glycerol, 50 mM KCl, 5 mM MgCl_2_, and 10 mM EDTA) with or without MYB62-MBP or AGB1-GST fused protein. For unlabeled probe competition, an unlabeled probe was added to the reaction, and single GST and MBP tags were used as negative controls. The probe sequence is shown in Appendix A.

### 2.12. Low Phosphate-Tolerant Phenotypic Assay of Plants

Sterilized *Arabidopsis* seeds were placed in a 1/2 MS medium and cultured in an incubator (24 °C/16 h, 20 °C/8 h). The seedlings were transplanted into a 1/2 MS medium (Caissonabs, Rexburg, ID, USA) and a phosphate-free medium (Caissonabs, USA); the media’s formulations are shown in Appendix A, respectively. The seedlings were then cultured in an incubator (24 °C/16 h, 20 °C/8 h) for 7–10 days.

## 3. Results

### 3.1. AGB1 Mutants agb1-2 and N692967 Were Dwarfed Compared with the WT after Exogenous Application of GA_3_

After treatment with 10 μΜ GA_3_, we found that the plant height of *agb1-2* was lower than that of the WT (Appendix A). In addition, we identified another homozygous *AGB1* mutant, *N692967* (SALK_204268C) (Appendix A), and carried out phenotypic experiments under te conditions of 1 μΜ GA_3_, 10 μΜ GA_3_, or 100 μΜ GA_3_ treatment (Appendix A). Without GA_3_ treatment, plant height and the rosette leaf of the mutants *agb1-2* and *N692967* were slightly smaller than in the WT (Figure 1A,B). When treated with different concentrations of GA_3_ for 10 days, the plant height of the mutants *agb1-2* and *N692967* was significantly lower than that of the WT (Figure 1D,E and Appendix A). Moreover, we found that following an increase in GA_3_ concentration, the plant height of *AGB1* mutant partially recovered compared with the WT, indicating that *AGB1* may be involved in the GA synthesis or degradation pathway. Therefore, in order to analyze the downstream pathway of *AGB1*, we analyzed the endogenous GA content of the *AGB1* mutant. The results showed that without GA_3_ treatment, there were no significant differences in GA content between the mutants *agb1-2* and *N692967*, and the WT (Figure 1C), but under GA_3_ treatment, the GA contents of *agb1-2* and *N692967* were significantly lower than that of the WT (Figure 1F), consistent with the results of their height (Figure 1B,E).

### 3.2. AGB1 Interacts with the DNA-Binding Region of MYB62, a GA Pathway Suppressor

Overexpression of *MYB62* reduced the sensitivity of plants to GA treatment and reduced tolerance to low-phosphorus stress [32] (Devaiah, Madhuvanthi, Karthikeyan, & Raghothama, 2009). This phenotype, under GA_3_ treatment and low-phosphorus stress, was similar to that of the *agb1-2* mutant in our study (Appendix A). Therefore, we tried to identify the interaction between AGB1 and MYB62 using a yeast two-hybrid assay. The yeast cells grew on selective media (SD/-Trp/-Leu/-His/-Ade) and selective media plus x-α-gels only when BD-AGB1 and AD-MYB62 fused proteins were co-expressed in yeast cells (Figure 2A). The pull-down experiment demonstrated that AGB1 and MYB62 interacted in vitro (Figure 2B). The firefly luciferase (LUC) complementary imaging (Lci) analysis demonstrated that when AGB1-nLUC and MYB62-cLUC were expressed in the leaves of *N. benthamiana*, strong LUC activity was observed, whereas there was no LUC activity in the negative control (including nLUC + cLUC, AGB1-nLUC + cLUC, and nLUC + MYB62-cLUC) (Figure 2C). The quantitative analysis results of LUC activity (Figure 2D) were consistent with those shown in Figure 2C. These results demonstrated that AGB1 can interact with MYB62 in plant cells.

To identify the interaction regions of MYB62, we inserted three truncated (M1–M3) segments as well as full-length *MYB62* into the AD vectors and inserted *AGB1* into the BD vector. These vectors were transformed into yeast cells to identify the interactions between different regions of *MYB62* and *AGB1*. Our results demonstrated that full-length *MYB62* interacted with AGB1, and the M1 and M2 truncated structures interacted slightly with AGB1, while M3 did not interact with AGB1 (Figure 3A), which suggests that the two DNA-binding regions of MYB62 were necessary for the interaction with AGB1.

In addition, in seedlings of WT *Arabidopsis*, the expression of *AGB1* was induced under GA_3_ treatment (Figure 3B), whereas the expression of *MYB62* was inhibited under GA_3_ treatment (Figure 3C). The tissue-specific expression of *AGB1* and *MYB62* at the seedling stage was analyzed in *Arabidopsis* (*Col-0*) without GA_3_ treatment. The results showed that *MYB62* was mainly expressed in *Arabidopsis* stems, which was similar to *AGB1* (Figure 3D,E). These results indicate that both *AGB1* and *MYB62* play an important role in stem development.

### 3.3. Genetic Analysis Indicated that MYB62 Was Downstream of AGB1 in the GA Pathway

In previous studies, researchers hoped to obtain homozygous mutants of *MYB62* by T-DNA insertion and antisense and RNAi-mediated *MYB62* silencing, but were unsuccessful and thus created the *MYB62* overexpression plant [32] (Devaiah, Madhuvanthi, Karthikeyan, & Raghothama, 2009). In order to study the genetic relationship between *AGB1* and *MYB62* further, we separately overexpressed *MYB62* in WT *Arabidopsis* (*MYB62:GFP/WT-8* and *MYB62:GFP/WT-10*) and *agb1-2* (*MYB62:GFP/agb1-2-1* and *MYB62:GFP/agb1-2-4*). Phenotypic analyses were carried out under the conditions of 1 μΜ GA_3_, 10 μΜ GA_3_, and 100 μΜ GA_3_, and the plant height was recorded (Figure 4 and Appendix A). Phenotypic analysis showed that without GA_3_ treatment, the mutants *agb1-2* and *N692967*, and the transgenic plants *MYB62:GFP/WT-8*, *MYB62:GFP/WT-10*, *MYB62:GFP/agb1-2-1*, and *MYB62:GFP/agb1-2-4* were slightly smaller than the WT (Figure 4A). The GA content of *MYB62:GFP/WT-8*, *MYB62:GFP/WT-10*, *MYB62:GFP/agb1-2-1*, and *MYB62:GFP/agb1-2-4* was significantly lower than that of the WT and *AGB1* mutants (Figure 4C).

Under 10 μM GA_3_ treatment, the height and the flowering time of the mutants *agb1-2* and *N692967* and the transgenic *Arabidopsis*, including *MYB62:GFP/WT-8*, *MYB62:GFP/WT-10*, *MYB62:GFP/agb1-2-1*, and *MYB62:GFP/agb1-2-4*, were significantly lower than in the WT (Figure 4D,E). The GA contents of the mutants *agb-1-2* and *N692967* and transgenic *Arabidopsis,* including *MYB62:GFP/WT-8*, *MYB62:GFP/WT-10*, *MYB62:GFP/agb1-2-1*, and *MYB62:GFP/agb1-2-4* were lower than that of the WT (Figure 4F), consistent with their height results (Figure 4D,E). In addition, the plant height of *MYB62:GFP/agb1-2* transgenic plants were more similar to that of *MYB62:GFP/WT*, and was higher than in the mutants *agb-1-2* and *N692967*, also consistent with their GA content results (Figure 4D–F). The height and GA content analysis was carried out under 1 μΜ GA_3_ and 100 μΜ GA_3_, and the results were consistent with those under 10 μΜ GA_3_ (Appendix A). The genetic analysis showed that *Arabidopsis MYB62* and *AGB1* belong to the same GA pathway, and *MYB62* is downstream of *AGB1* in this GA pathway.

### 3.4. The Interaction between AGB1 and MYB62 Did Not Affect the Transcriptional Activation of MYB62

The subcellular localization of MYB62 protein was completed in WT *Col-0 Arabidopsis* protoplasts. The results showed that green fluorescent protein (GFP) was expressed in the nucleus and cell membrane of the protoplasts transformed with the vector 16318-AGB1-GFP (Figure 5A), indicating that the AGB1 protein was located in the nucleus and cell membrane. GFP was expressed in the nucleus of the protoplasts transformed with the vector 16318-MYB62-GFP (Figure 5A), consistent with a previous report on the localization of MYB62 in plant cells [32] (Devaiah, Madhuvanthi, Karthikeyan, & Raghothama, 2009). These results indicated that AGB1 and MYB62 are co-located in the nucleus.

In order to further explore whether *AGB1* affects the transcriptional activity of *MYB62*, we carried out transcriptional activation experiments in yeast cells (Figure 5B,C). We constructed a pBridge-MYB62 vector to detect the transcriptional activation activity of *MYB62* in yeast cells. In addition, a pBridge-MYB62-AGB1 vector was constructed by inserting *AGB1* into pBridge-MYB62, and the effect of *AGB1* on *MYB62* transcriptional activation was detected (Figure 5B). The results showed that the yeast transformed with the pBridge-MYB62 vector grew on the selective medium (Figure 5C). The growth of the yeast transformed with pBridge-MYB62-AGB1 was similar to that of pBridge-MYB62 (Figure 5C), indicating that *MYB62* had transcriptional activation activity, and *AGB1* had no effect on the transcriptional activation of *MYB62* in yeast cells.

### 3.5. MYB62 Can Bind the Promoter of GA Degradation Gene GA2ox7 to Enhance Its Expression, and AGB1 Negatively Regulates the DNA-Binding Activity of MYB62 on the Promoter of GA2ox7

*AGB1* did not affect the transcriptional activation of *MYB62*. Therefore, we speculated that *AGB1* may affect the DNA binding activity of *MYB62*. Thus, we analyzed the expression of many genes related to GA synthesis and degradation in the WT, *agb1-2*, and *MYB62:GFP/WT-10* under GA_3_ treatment. We found that the *GA2ox7* gene (At1g47990), which is related to the degradation process of GA, may be related to the regulation pathway of AGB1-MYB62. Gene expression analysis showed that the expression of *GA2ox7* in the WT was significantly lower than that in the mutants *agb1-2* and *N692967*, and transgenic plants, including *MYB62:GFP/WT-8*, *MYB62:GFP/WT-10*, *MYB62:GFP/agb1-2-1*, and *MYB62:GFP/agb1-2-4* (Figure 6A). This result was consistent with the GA content results (Figure 4F), suggesting that *GA2ox7* is regulated by AGB1-MYB62. Moreover, we found that the expression of *GA2ox7* in *MYB62:GFP/agb1-2-1* and *MYB62:GFP/agb1-2-4* was similar to that in *MYB62:GFP/WT-8* and *MYB62:GFP/WT-10*, but lower than in the mutants *agb1-2* and *N692967* (Figure 6A), also consistent with the GA content results (Figure 4F). In addition, EMSA analysis and LUC (luciferase analysis) showed that *MYB62* bound to the MYB element (TGGTTG) in the *GA2ox7* promoter and enhanced the expression of *GA2ox7* (Figure 6B–F). However, *AGB1* can negatively regulate the binding of *MYB62* to the *GA2ox7* promoter, thus negatively regulating the expression of the *GA2ox7* promoter in plants (Figure 6G–I). These results indicated that AGB1 interacted with the DNA-binding region of MYB62, further negatively regulating MYB62 binding on the downstream gene *GA2ox7*, and actively participating in the GA pathway.

## 4. Discussion

### 4.1. AGB1 Regulates the GA Pathway by Negatively Regulating MYB62 Activity

In this study, we demonstrated the molecular mechanism of how *AGB1* regulates the GA pathway (Figure 7). We found that *AGB1* was upregulated and *MYB62* was downregulated under GA_3_ treatment (Figure 3B,C). Both *AGB1* and *MYB62* were mainly expressed in the stem of *Arabidopsis* (Figure 3D,E). Through phenotype analysis of the mutant *agb1* and *MYB62*-overexpressing plants under GA_3_ treatment, *AGB1* was found to positively regulate the response to the GA pathway, and *MYB62* negatively regulated the response to the GA pathway (Figure 4). Genetic analysis showed that *AGB1* and *MYB62* were involved in the same GA pathway, and that *MYB62* was downstream (Figure 4). Biochemical experiments showed that AGB1 interacted with the DNA-binding region of MYB62, and that MYB62 could directly bind to the promoter of the GA degradation gene *GA2ox7* and promote its expression (Figure 3 and Figure 6). AGB1 inhibited the binding of MYB62 on the *GA2ox7* promoter, thus negatively regulating the activity of MYB62 and positively regulating the GA pathway in *Arabidopsis* (Figure 6G–I).

We found that the interaction between AGB1 and MYB62 did not affect the transcriptional activation of *MYB62* (Figure 5C). AGB1 interacted with the DNA-binding region of MYB62 (Figure 3A) and negatively regulated the activity of MYB62 by affecting binding to the promoter region of the metabolic gene *GA2ox7* (Figure 6G–I), though neither affected the transcription activity of *MYB62*. Our previous research on *AGB1* in *Arabidopsis* determined that *AGB1* inhibits the transcriptional activation activity of *BBX21*, a positive regulatory factor, by binding to the transcriptional activation region of *BBX21*, thereby regulating the expression of the downstream genes [17] (Xu et al., 2017). Taken together, these results suggest that *AGB1* regulates the activity of downstream transcription factors by combining different regions of the downstream transcription factors. Other research has found that the G protein β subunit regulates plant hypocotyl elongation [17] (Xu et al., 2017), BR signal transduction [37] (Zhang et al., 2018), and plant nutrition regulation [38] (Wu et al., 2020) by interacting with different transcription factors. These results suggest that *AGB1* binding with different downstream transcription factors may be a general mechanism to regulate different signaling pathways in *Arabidopsis*.

In *Arabidopsis*, the G protein α subunit *GPA1* has an important role in regulating hypocotyl elongation, ABA inhibition of the stomatal opening, stomatal density, pollen tube development, and plant height, but the most common regulation mode is the GPA1–GCR1 coupling complex [39] (Chakraborty, Singh, Kaur, & Raghuram, 2015). In rice, the G-α subunit positively regulates the GA pathway by inhibiting the activity of a negative regulator, SLR (slender rice), in GA signaling [13] (Ueguchi-Tanaka et al., 2000). These studies suggested that different subunits of the G protein complex can positively regulate the GA pathway through different downstream genes in plants.

### 4.2. The AGB1–MYB62 Pair Is Involved in Regulating the Phosphate Starvation Response in Plants

*MYB62* participates in the response to low-phosphorus stress by negatively regulating the synthesis of GA [32] (Devaiah, Madhuvanthi, Karthikeyan, & Raghothama, 2009). Therefore, in order to study whether AGB1–MYB62 also functions in phosphorus deficiency, we analyzed the phenotypes of the WT, the *agb1-2* mutant, *MYB62-GFP/WT-10*, and *MYB62-GFP/agb1-2-4* under phosphorus-free conditions (Appendix A). The results showed that under normal conditions, the length of the main root and the number of lateral roots of *MYB62:GFP/WT-10* were lower than those of the WT (Appendix A). This is consistent with the results of Devaiah et al. (2009). In this study, under normal conditions, there was no significant difference in main root length and lateral root number between *agb1-2* and the WT. Under phosphorus-free conditions, the main root length and lateral root number of *agb1-2* were significantly lower than those of the WT, while those of *MYB62-GFP/agb1-2-4* were significantly lower than those of the WT and consistent with *MYB62-GFP/WT-10* (Appendix A). These results indicate that *AGB1* positively regulated the response to phosphorus starvation in plants by stimulating root growth. The AGB1–MYB62 pair is also involved in regulating the root growth process under phosphorus starvation. More evidence is needed to demonstrate how the AGB1–MYB62 pair coordinates the regulation of GA and low-phosphorus stress responses in plants.

## Figures and Tables

**Figure 1 ijms-22-08270-f001:**
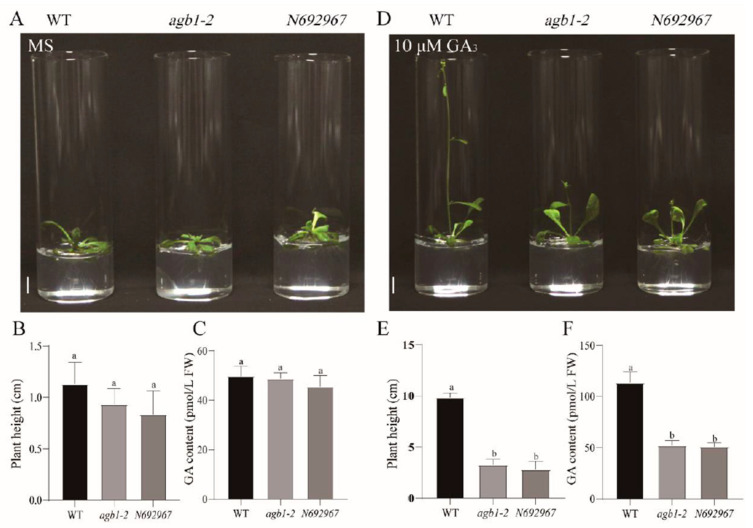
Phenotypes of the wild-type (Col-0), *agb1-2*, and *N692967* under GA_3_ treatment. (**A**) Plant height phenotypes of the wild-type (Col-0), *agb1-2*, and *N692967* under normal conditions. Scale bars, 1 cm. (**B**,**C**) Plant height and GA content of the wild-type (Col-0), *agb1-2*, and *N692967* under normal conditions. Data are the average of three independent experiments, and the error bars represent SE (*n* = 10). Significant differences were analyzed using Duncan’s multiple range test (*p* < 0.05). (**D**) Plant height phenotypes of the wild-type (Col-0), *agb1-2*, and *N692967* treated with 10 μM GA_3_. Scale bars, 1 cm. (**E**,**F**) Plant height and GA content of the wild-type (Col-0), *agb1-2*, and *N692967* under normal conditions. Data are the average of three independent experiments, and the error bar represents the SE (*n* = 10). Significant differences were analyzed using Duncan’s multiple range test (*p* < 0.05).

**Figure 2 ijms-22-08270-f002:**
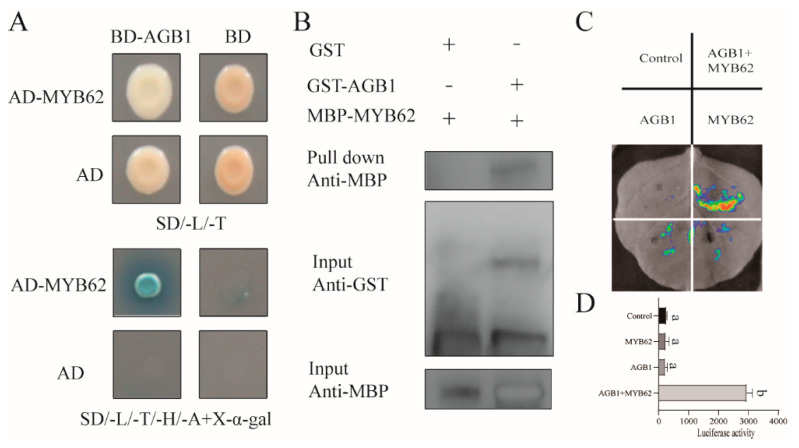
The interaction between AGB1 and MYB62. (**A**) Yeast two-hybrid interactions between full-length sequences of *AGB1* and *MYB62*. The transformed yeast cells were activated and cultured on SD/-Leu/-Trp and SD/-Leu/-Trp/-His/-Ade media. The yeast cells of the empty vectors AD and BD were used as negative controls. (**B**) Protein interaction of AGB1 and MYB62. In vitro GST pull-down assays showed that AGB1 interacted with MYB62 in vitro. (**C**) Interaction of AGB1 with MYB62. A luciferase (LUC) assay was performed to demonstrate that AGB1 and MYB62 can interact with *N. benthamiana* leaf cells. AGB1 and MYB62 were fused with nLUC and cLUC, respectively. nLUC-only and cLUC-only refer to the empty vectors used as negative controls. (**D**) Quantification of luminous intensity in C. Error bars represent the means ± SE (*n* = 3). Significant differences were analyzed using Duncan’s multiple range test (*p* < 0.05).

**Figure 3 ijms-22-08270-f003:**
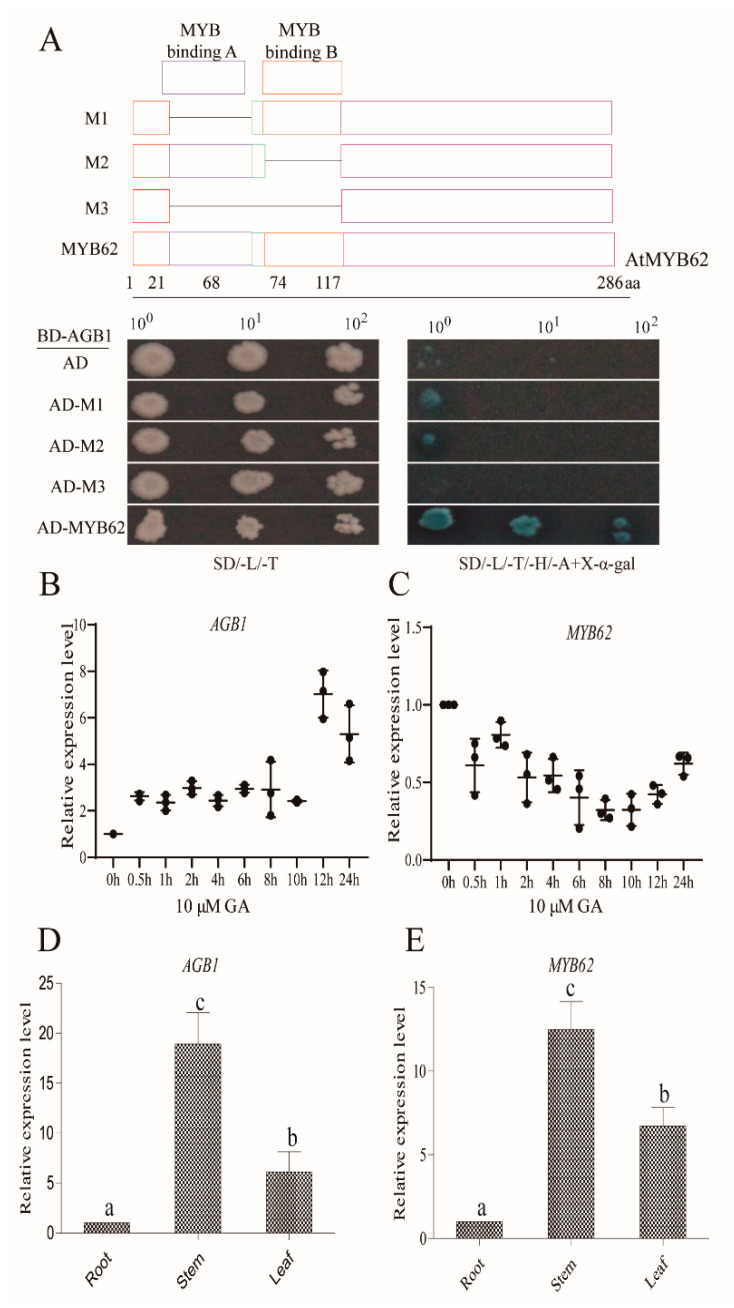
*AGB1* and *MYB62* expression analysis. (**A**) Yeast two-hybrid interactions between the full-length sequence of AGB1 and the piecewise sequence of MYB62. The transformed yeast cells were activated and cultured on SD/-Leu/-Trp and SD/-Leu/-Trp/-His/-Ade media. The yeast cells of the empty vectors AD and BD were used as negative controls. (**B**) Analysis of AGB1 expression in wild-type Col-0 leaves grown for 4 weeks after treatment with 10 μM GA3 at different time periods. The data are the average of three independent experiments, and the error bar represents the SE (*n* = 3). Significant differences were analyzed using Duncan’s multiple range test (*p* < 0.05). Relative quantitative results were calculated by normalization using the control gene ACT2 (AT3G18780). (**C**) Analysis of MYB62 expression in wild-type Col-0 leaves, grown for 4 weeks after treatment with 10 μM GA3 at different time periods. The data are the average of three independent experiments, and the error bar represents the SE (*n* = 3). Significant differences were analyzed using Duncan’s multiple range test (*p* < 0.05). (**D**) Analysis of AGB1 expression in the roots, stem, and leaves of wild-type Col-0 grown for 4 weeks under normal growth conditions. The data are the average of three independent experiments, and the error bar represents the SE (*n* = 3). a, b c indicate significant differences. Significant differences were analyzed using Duncan’s multiple range test (*p* < 0.05). (**E**) Analysis of MYB62 expression in the roots, stem, and leaves of wild-type Col-0 grown for 4 weeks under normal growth conditions. The data are the average of three independent experiments, and the error bar represents the SE (*n* = 3). Significant differences were analyzed using Duncan’s multiple range test (*p* < 0.05).

**Figure 4 ijms-22-08270-f004:**
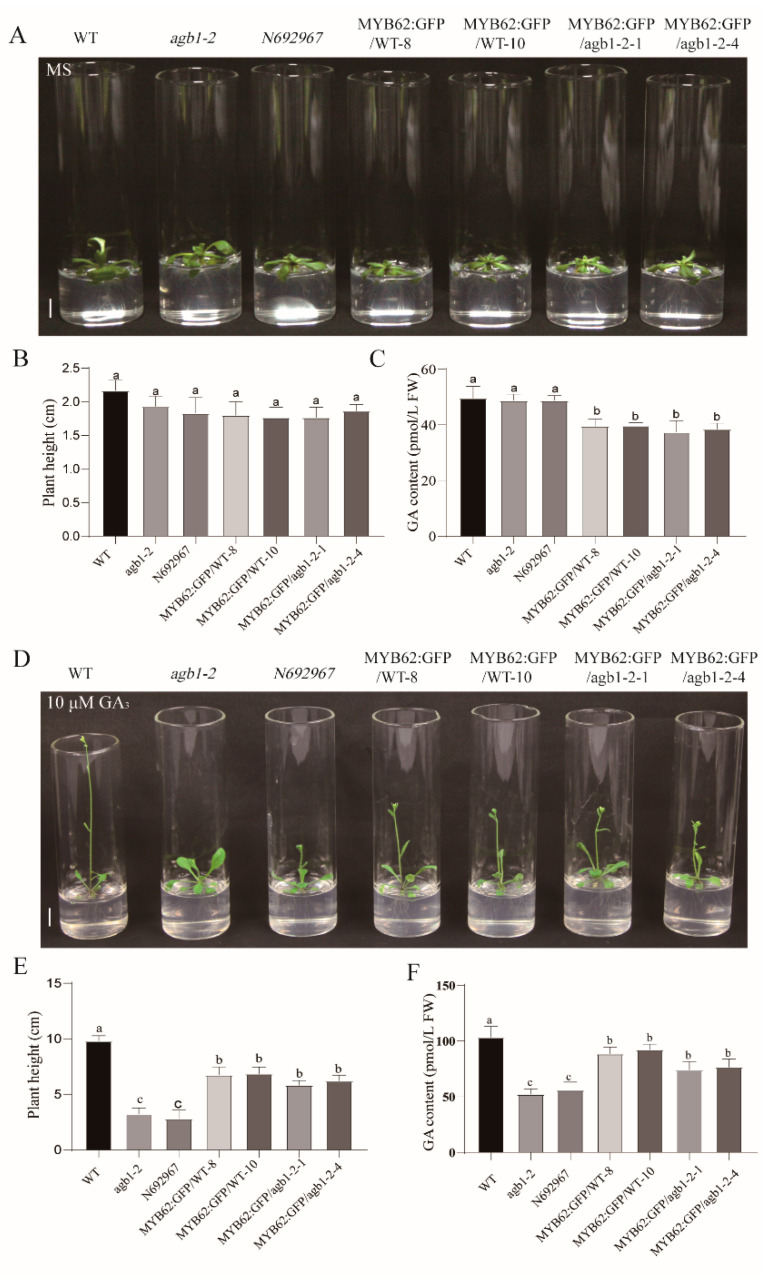
Phenotypic identification of the wild-type, *agb-1-2*, *N692967*, *MYB62*:GFP/WT-8, *MYB62*:GFP/WT-10, *MYB6*2:GFP/*agb1-2*-*1*, and *MYB62*:GFP/*agb1-2*-*4* under normal conditions and 10 μM GA_3_ treatment. (**A**) Plant height phenotype of the wild-type (Col-0), *agb-1-2*, *N692967*, *MYB62*:GFP/WT-8, *MYB62*:GFP/WT-10, *MYB62*:GFP/*agb1-2-1*, and *MYB62*:GFP/*agb1-2-4* under normal conditions. Scale bars, 1 cm. (**B**,**C**) Plant height and GA content of the wild-type (Col-0), *agb-1-2*, *N692967*, *MYB62*:GFP/WT-8, *MYB62*:GFP/WT-10, *MYB62*:GFP/*agb1-2-1*, and *MYB62*:GFP/*agb1-2-4* under normal conditions. The data are the average of three independent experiments, and the error bar represents the SE (*n* = 10). a, b indicate significant differences. Significant differences were analyzed using Duncan’s multiple range test (*p* < 0.05). (**D**) Plant height phenotype of the wild-type (Col-0), *agb-1-2*, *N692967*, *MYB62*:GFP/WT-8, *MYB62*:GFP/WT-10, *MYB62*:GFP/*agb1-2-1*, and *MYB62*:GFP/*agb1-2-4* under 10 μΜ GA_3_ treatment. Scale bars, 1 cm. (**E**,**F**) Plant height and GA content of the wild-type (Col-0), *agb-1-2*, *N692967*, *MYB62*:GFP/WT-8, *MYB62*:GFP/WT-10, *MYB62*:GFP/*agb1-2-1*, and *MYB62*:GFP/*agb1-2-4* under 10 μΜ GA_3_ treatment. The data are the average of three independent experiments, and the error bar represents the SE (*n* = 10). a, b c indicate significant differences. Significant differences were analyzed using Duncan’s multiple range test (*p* < 0.05).

**Figure 5 ijms-22-08270-f005:**
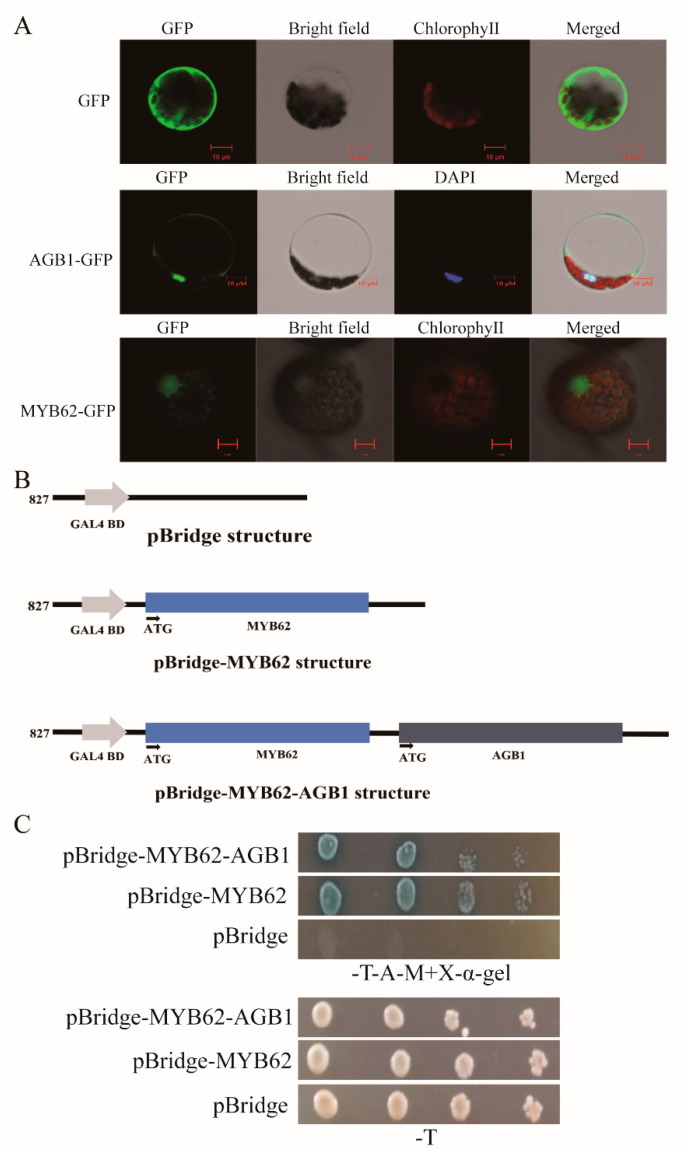
The interaction between AGB1 and MYB62 did not affect the transcriptional activation of MYB62; AGB1 and MYB62 are both located in the nucleus. (**A**) Analysis of the subcellular localization of AGB1 and MYB62 in the protoplasts of wild-type *Arabidopsis Col-0* leaves grown for 3 weeks. Scale bars, 10 μm. (**B**) The structure of the pBridge vector, pBridge-MYB62, and pBridge-MYB62-AGB1. (**C**) The vectors pBridge, pBridge-MYB62, and pBridgeMYB62-AGB1 were transferred into AH109 yeast cells, and the cells were diluted and transferred to selective media (SD/-Trp, SD/-Trp-Ade, SD/-Trp-His-Ade) to observe their growth.

**Figure 6 ijms-22-08270-f006:**
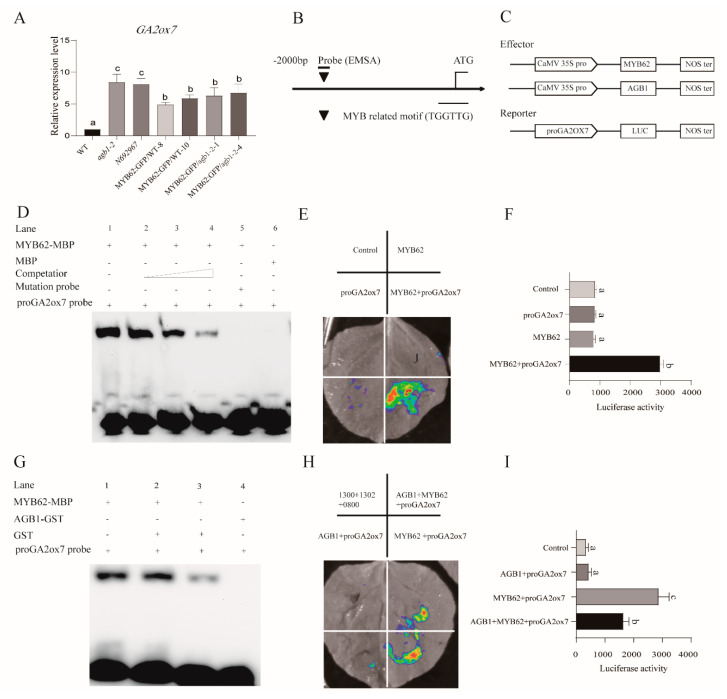
*AGB1* negatively regulates the expression of the downstream genes of *MYB62*. (**A**) The GA degradation-related gene, *GA2ox7*, in the wild-type (*Col-0*), *agb-1-2*, *N692967*, *MYB62*:GFP/WT-8, *MYB62*:GFP/WT-10, *MYB62*:GFP/*agb1-2*-*1*, and *MYB62:GFP/agb1-2*-*4* revealed by qRT-PCR analysis under GA_3_ treatment. Error bars represent the means ± SE *(n* = 3). Significant differences were analyzed using Duncan’s multiple range test (*p* < 0.05). Relative quantitative results were calculated by normalization using the control gene *ACT2* (AT3G18780). (**B**) Illustration of the MYB62 promoter region showing the presence of the MYB-binding site. (**C**) Schematic diagram of the effectors and the reporter structure of LUC (luciferase analysis) in E and H. (**D**) The EMSA (electrophoretic mobility shift assay) experiment showed that *MYB62* could bind to the promoter of *GA2ox7*. (**E**) The LUC experiment showed that *MYB62* could bind to the *GA2**ox7* promoter and promote the expression of *GA2ox7*; 302 indicates the pCambia 1302 vector, and 0800 indicates the pGreenII 0800-LUC vector. Representative images of *N. benthamiana* leaves were taken 48 h after infiltration. All experiments were repeated three times with similar results. (**F**) Quantification of luminescence intensity in E. Error bars represent the means ± SE *(n* = 3). Significant differences were analyzed using Duncan’s multiple range test (*p* < 0.05). (**G**) The EMSA experiment showed that *AGB1* negatively regulates the binding of *MYB62* to the e*GA2ox7* promoter. (**H**) The LUC experiment showed that *AGB1* could negatively regulate the binding of *MYB62* to the *GA2ox7* promoter and negative regulately the expression of *GA2ox7*; 1300 indicates the pCambia 1300 vector, 1302 indicates the pCambia 1302 vector, and 0800 indicates the pGreenII 0800-LUC vector. Representative images of *N. benthamiana* leaves were taken 48 h after infiltration. All experiments were repeated three times with similar results. (**I**) Quantification of luminescence intensity in H. Error bars represent the means ± SE (*n* = 3). Significant differences were analyzed using Duncan’s multiple range test (*p* < 0.05).

**Figure 7 ijms-22-08270-f007:**
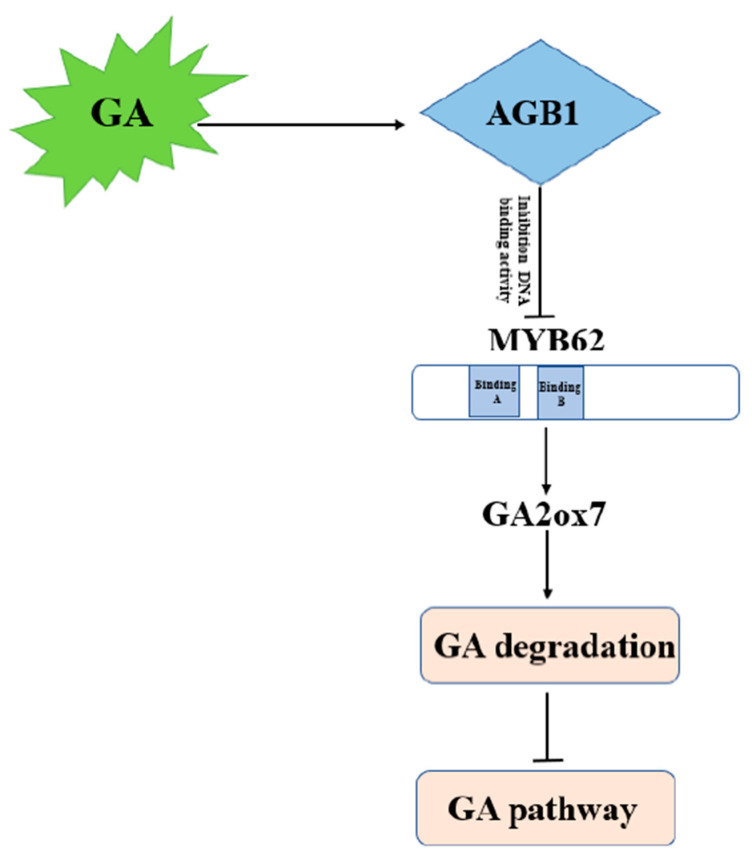
Model of how AGB1 regulates GA-related signaling pathways by controlling MYB62. This regulatory model shows the role of AGB1–MYB62 in regulation of the GA signaling pathway in *Arabidopsis thaliana*. AGB1 positively participates in the GA signaling pathway, interacts with the negative transcription factor MYB62, inhibits the binding of MYB62 to the promoter of GA metabolism related gene *GA2ox7*, and negative regulates *GA2ox7* expression.

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
