# Peer review of "Arabidopsis G-Protein β Subunit AGB1 Negatively Regulates DNA Binding of MYB62, a Suppressor in the Gibberellin Pathway"

_ijms, 2021, doi:10.3390/ijms22158270_

Round 1

Reviewer 1 Report

In their manuscript, Qi et al. show that G protein β subunit (AGB1) interacts with MYB62 involved in GA pathway, and that this interaction inhibits the binding of MYB62 to the GA2ox7 promoter. While the subject is overall interesting and the interaction not reported previously, the results do not provide a deep analysis of AGB1-MYB62 interrelation, and the manuscripty conteins much inconsistency, which makes it too preliminary. Also, the language needs extensive editing as it currently contains a lot of mistakes. Detailed comments are listed below:

1) some interaction data present very weak signals or are of low quality (pull-down, Fig. 2B, Y2H, Fig. 3A). It is not clear whether C-terminal parts of the protein were included in M1 and M2. Does M1+M2 fragment show strong interaction?

2) Fig.1 and Fig.4 : the investigated phenotype upon GA treatment is described as changes in plant heights, while it rather shows the impact of gibberellins on flowering/bolting (the earlier flowering plants simply have higher stems compared to later flowering ones). 

3) Fig. 5A: the fluorescence signals of MYB62-GFP and AGB1 -GFP seem different, and based on the pictures shown I would not be convinced that AGB1 -GFP has nuclear localisation. Also, the statement in Results section that MYB62-GFP and AGB1 -GFP co-localize is not supported by the data. 

3) the binding of MYB62 to GA2ox7 promoter sequence is interesting but does it occur in vivo ?  

4) there are repetitions in the manuscript, for example the fragment about difficulties in obtaining myb62 mutant is mentioned several times. In addition, some parts of the text are not clear or have mistakes, for example Fig. 5 title : `The interaction between AGB1 and MYB62 did not affect the transcriptional activation of MYB62, and AGB1 and MYB62 are both located in the nucleus`

5) in many places in the manuscript published data are described without reference

Author Response

Reviewer 1

In their manuscript, Qi et al. show that G protein β subunit (AGB1) interacts with MYB62 involved in GA pathway, and that this interaction inhibits the binding of MYB62 to the GA2ox7 promoter. While the subject is overall interesting and the interaction not reported previously, the results do not provide a deep analysis of AGB1-MYB62 interrelation, and the manuscript contain much inconsistency, which makes it too preliminary. Also, the language needs extensive editing as it currently contains a lot of mistakes. Detailed comments are listed below:

1) some interaction data present very weak signals or are of low quality (pull-down, Fig. 2B, Y2H, Fig. 3A).

Answer:

Thanks for your advice. In this manuscript, we used three methods to demonstrate the AGB1-MYB62 interaction, including the Y2H (Fig. 2A), the pull-down (Fig. 2B), and the Lci (Fig. 2C), and all experimental results of three methods were consistent. For the Pull-down results (Fig. 2B), because the background of the original picture was too black, the brightness of the picture was turned too bright. In the revised manuscript, according to the reviewers' opinions, we dimmed the background of the picture and improve the bands clearer (line 427). For the Y2H results showed in Fig. 3A, we also adjusted the brightness of the picture according to the opinions of the reviewers, and the picture became clearer (line 437).

It is not clear whether C-terminal parts of the protein were included in M1 and M2. Does M1+M2 fragment show strong interaction.

Answer:

Both M1 and M2 truncated protein includes C-terminal parts of MYB62, and Fig. 3A showed the schematic diagram of the structure of MYB62. We did not mix M1+M2 fragments to detect interactions with AGB1. We have identified that full-length MYB62 can interact with AGB1 (Fig 2A, 3A), and absence of both MYB domains greatly reduces the interaction, and the absence of a single MYB domain partly reduces the interaction between MYB62 and AGB1(Fig 3A). These results demonstrate that the both MYB domains of MYB62 are important for the interaction with AGB1.

2) Fig.1 and Fig.4: the investigated phenotype upon GA treatment is described as changes in plant heights, while it rather shows the impact of gibberellins on flowering/bolting (the earlier flowering plants simply have higher stems compared to later flowering ones).

Answer:

Thank the reviewers for their very good suggestions. In fact, we compared flowering time between the different plants and found that the agb1 mutants and MYB62 overexpression plants, including agb1-2, N692967, MYB62:GFP/WT-8, MYB62:GFP/WT-10, MYB62:GFP/agb1-2-1, MYB62:GFP/agb1-2-4, flowered later than WT, which were consistent with their plant height results. In the old manuscript, because the phenotype of plant height is easy to observe in the picture, we used plant height as the main phenotypic character, and did not describe the flowering time character. In the revised manuscript, according to the reviewer's suggestion, we added the phenotypic description of flowering time to the experimental result part (line307).

3) Fig. 5A: the fluorescence signals of MYB62-GFP and AGB1 -GFP seem different, and based on the pictures shown I would not be convinced that AGB1 -GFP has nuclear localisation. Also, the statement in Results section that MYB62-GFP and AGB1 -GFP co-localize is not supported by the data.

Answer:

We believe that MYB62-GFP and AGB1-GFP are indeed differently positioned. MYB62-GFP is localized in the nucleus, which is consistent with results in published reference (Davidson, S. E. Elliott, R. C. Helliwell, C. A. Poole, A. T, Reid, J. B. (2003). The pea gene NA encodes ent-kaurenoic acid oxidase. Plant Physiol, 131(1), 335-344, doi: 10.1104/pp.012963). In this research, we found that AGB1 localized in the cell membrane and nucleus, and AGB1 nuclear localization can be identified by the co-localization of DIP nuclear specific dye (Fig. 5A). In fact, except us, other researchers have also found that AGB1 can be localized in the nucleus (Botella J. (2007). Expression analysis and subcellular localization of the Arabidopsis thaliana G-protein β-subunit AGB1. Plant Cell Rep, 26:1469–1480, doi: 10.1007/s00299-007-0356-1). Wether AGB1 is localized in the nucleus remains to be further investigated.

4) the binding of MYB62 to GA2ox7 promoter sequence is interesting but does it occur in vivo?

Answer:

Thanks for your question. We have identified the binding of MYB62 to GA2ox7 promoter sequence in vitro using the EMSA method (Fig. 6D, 6G). In fact, we have no evidence to identify direct binding of MYB62 to GA2ox7 promoter in vivo, but we demonstrated that MYB62 can activate the activity of the GA2ox7 promoter in tobacco by LUC assay (Figure 6E, 6F, 6H, 6I).

5) there are repetitions in the manuscript, for example the fragment about difficulties in obtaining myb62 mutant is mentioned several times. In addition, some parts of the text are not clear or have mistakes, for example Fig. 5 title : `The interaction between AGB1 and MYB62 did not affect the transcriptional activation of MYB62, and AGB1 and MYB62 are both located in the nucleus`

Answer:

Thanks for reviewer comments. We have revised manuscript according to the reviewer's suggestion (line323).

6) in many places in the manuscript published data are described without reference

Answer:

Thanks for reviewer comments. We have added references in revised manuscript (line 38, line 43-45, line 529).

Reviewer 2 Report

Heterotrimeric G-protein signaling exhibits pleiotropic effects on development of plant organism. Most of effects observed on mutants or transgenic plants were connected somehow to elongation of stem or other organs. This could indicate important crosstalk of G-protein signaling with GA3 perception or regulation. Surprisingly, there are very few works describing connection of G-protein signaling and gibberellin sensing. Therefore, I consider submitted manuscript as highly interesting and innovative. I also appreciate multidirectional approach to testing of presented hypotheses.

I have several comments to the manuscript:

1) Lines 43-45: Numerous effects of G-protein signaling on plant development are listed and only one source cited (Wang et al., 2011). All primary authors of each finding should be cited.

2) There are in two paragraphs (lines 51-65; 80-91) described specific regulatory mechanisms involving G-protein signaling. Each mechanism is explained by well-constructed story of original findings and molecular mechanisms. The problem I can see is mixing the original findings observed on Eudicots as well as Poales. Accorging to current knowledge (see e.g. review by Pandey 2019), it seems that G-protein signaling in Poales works in very alternative way to all the knowledge we have from Eudicots (especially Arabidopsis thaliana). I consider building the hypothesis on mixing knowledge from Poales and Eudicots as risky.

3) Paragraph describing cellular localization of GFP constructs (lines 325-327). The localization of AGB1:GFP is reported in nuclei as well as in plasma membrane. Do authors have any evidence the nuclear localization is specific? Did authors consider possibility of accumulation of free GFP in nuclei as a consequence of degradation of the overexpressed fusion protein?

Author Response

Reviewer 2

Comments and Suggestions for Authors

Heterotrimeric G-protein signaling exhibits pleiotropic effects on development of plant organism. Most of effects observed on mutants or transgenic plants were connected somehow to elongation of stem or other organs. This could indicate important crosstalk of G-protein signaling with GA3 perception or regulation. Surprisingly, there are very few works describing connection of G-protein signaling and gibberellin sensing. Therefore, I consider submitted manuscript as highly interesting and innovative. I also appreciate multidirectional approach to testing of presented hypotheses.

I have several comments to the manuscript:

1) Lines 43-45: Numerous effects of G-protein signaling on plant development are listed and only one source cited (Wang et al., 2011). All primary authors of each finding should be cited.

Answer:

Thanks for reviewer comments. We have added references in revised manuscript (line 43-45, and line 529).

2) There are in two paragraphs (lines 51-65; 80-91) described specific regulatory mechanisms involving G-protein signaling. Each mechanism is explained by well-constructed story of original findings and molecular mechanisms. The problem I can see is mixing the original findings observed on Eudicots as well as Poales. Accorging to current knowledge (see e.g. review by Pandey 2019), it seems that G-protein signaling in Poales works in very alternative way to all the knowledge we have from Eudicots (especially Arabidopsis thaliana). I consider building the hypothesis on mixing knowledge from Poales and Eudicots as risky.

Answer:

Thanks for reviewer comments. Indeed, when we described the signaling pathway of G protein, we ignored the difference between Poales and Eudicots. Because we mainly revealed the mechanism of G protein in Arabidopsis, according to the reviewer's suggestion, we deleted the description of rice G protein (line 54).

3) Paragraph describing cellular localization of GFP constructs (lines 325-327). The localization of AGB1:GFP is reported in nuclei as well as in plasma membrane. Do authors have any evidence the nuclear localization is specific? Did authors consider possibility of accumulation of free GFP in nuclei as a consequence of degradation of the overexpressed fusion protein?

Answer:

Thanks for reviewer comments. In this research, we found that AGB1 localized in the cell membrane and nucleus, and AGB1 nuclear localization can be identified by the co-localization of DIP nuclear specific dye (Fig. 5A). In fact, except us, other researchers have also found that AGB1 can be localized in the nucleus (Botella J. (2007). Expression analysis and subcellular localization of the Arabidopsis thaliana G-protein β-subunit AGB1. Plant Cell Rep, 26:1469–1480, doi: 10.1007/s00299-007-0356-1).. We agreed that the nuclear localization of AGB1 is due to the degradation of the overexpressed fusion protein and the accumulation of free GFP in the nucleus. Whether AGB1 is localized in the nucleus remains to be further investigated

Reviewer 3 Report

The manuscript is quite interesting. I would just suggest that the authors used the words “inhibit” and “epistasis” cautiously. The phenotype and the results from the analyses using MYB62 overexpression in the wild type and agb1 backgrounds do not show an expressive difference for the use of these words (Figure 4 and 6A). Nonetheless, under GA treatment, it is clear that GA content has been degraded in the agb1 mutants (Figure 1F), which would indicate the "free" action of MYB62 within the GA2ox7 promoter, as the authors proposed here. However, I believe that the authors used leaf tissues for all the expression gene analysis in order to infer and understand the stem elongation phenotypes in the plants. Thus, it might be that stem elongation is regulated by GA within the rib zone meristem in a different manner from leaves ( as we observe for the genes as well; Figure 3D and E). I also noted, and the authors may correct me please, that under GA treatment, the agb1 mutants showed leaf hyponasty (Figure 1D, S2 and S4). Auxin is known to play this function in the leaf, indicating that what is happening in the leaf might indeed be different from the stem (mainly the rib zone region).   

Here are some suggestions and comments:

  • The author used standard deviation (SD) for inferential statistics. This type of error bar is used just for descriptive analysis. I suggest that the authors use either SE or 95% CI.
  • Bar graph is not appropriate for continuous data, only for percentage or counts. In case you have small sample size (n=3), the preferred option is to show individual data as scatter plot. However, in the case that the authors still want to keep bar graph, I strongly suggest at least overlaying individual point within the bar graph.
  • The authors mentioned that the data is the average of three independent experiments, but the error bar represents n= 10. This is a bit confused.

Author Response

Reviewer 3

Comments and Suggestions for Authors

The manuscript is quite interesting. I would just suggest that the authors used the words “inhibit” and “epistasis” cautiously. The phenotype and the results from the analyses using MYB62 overexpression in the wild type and agb1 backgrounds do not show an expressive difference for the use of these words (Figure 4 and 6A). Nonetheless, under GA treatment, it is clear that GA content has been degraded in the agb1 mutants (Figure 1F), which would indicate the "free" action of MYB62 within the GA2ox7 promoter, as the authors proposed here. However, I believe that the authors used leaf tissues for all the expression gene analysis in order to infer and understand the stem elongation phenotypes in the plants. Thus, it might be that stem elongation is regulated by GA within the rib zone meristem in a different manner from leaves (as we observe for the genes as well; Figure 3D and E). I also noted, and the authors may correct me please, that under GA treatment, the agb1 mutants showed leaf hyponasty (Figure 1D, S2 and S4). Auxin is known to play this function in the leaf, indicating that what is happening in the leaf might indeed be different from the stem (mainly the rib zone region).   

Answer:

Thanks for reviewer comments. According to your suggestion, I have replaced word “inhibit” with “negative regulate” (line 342, 358, 360, 361, 364), and replaced word “epistasis assay” with “genetic assay” (line 24, 296, 320, 373).

     We agree that the regulatory mechanism of GA in stem and leaf is different. In fact, when we analyzed the gene expression and GA content, we obtained the mixed samples of plant stem and leaf without separate sample. We also noticed the phenotype of agb1 mutant leaf hyponasty, which may be related to auxin hormone, indicating that the mechanism of agb1 regulating leaf development is different to the mechanism of regulating stem elongation. For this interesting phenotype, we need to separate the leaf and stem to analyze their regulatory mechanisms in the future.

Here are some suggestions and comments:

  • The author used standard deviation (SD) for inferential statistics. This type of error bar is used just for descriptive analysis. I suggest that the authors use either SE or 95% CI.

Answer:

Thanks for reviewer advices. We have modified it in revised manuscript.

  • Bar graph is not appropriate for continuous data, only for percentage or counts. In case you have small sample size (n=3), the preferred option is to show individual data as scatter plot. However, in the case that the authors still want to keep bar graph, I strongly suggest at least overlaying individual point within the bar graph.

Answer:

Thanks for reviewer advices. We have modified it in revised manuscript (Fig 3B and Fig 3C).

  • The authors mentioned that the data is the average of three independent experiments, but the error bar represents n= 10. This is a bit confused.

Answer:

Thanks for reviewer comments. Three independent biological experiments were performed in the luciferase analysis of MYB62 binding to the GA2ox7 promoter. We have modified it in revised manuscript.

Round 2

Reviewer 1 Report

While the Authors answered to some doubts concerning protein interaction analyses, the manuscript still suffers from limited physiological analyses which are prone to mis-interpretations. The manuscript still do not show convincingly what are the functions of the studied factors in regulating GA responses.